# Incremental Learning of Discrete Planning Domains from Continuous Perceptions

**Luciano Serafini**, **Paolo Traverso**
Fondazione Bruno Kessler
{serafini,traverso}@fbk.eu

paper adhering to the theme on **model acquisition**

## Abstract

We propose a framework for learning discrete deterministic planning domains. In this framework, an agent learns the domain by observing the action effects through continuous features that describe the state of the environment after the execution of each action. Besides, the agent learns its *perception function*, i.e., a probabilistic mapping between state variables and sensor data represented as a vector of continuous random variables called *perception variables*. We define an algorithm that updates the planning domain and the perception function by (i) introducing new states, either by extending the possible values of state variables, or by weakening their constraints; (ii) adapts the perception function to fit the observed data (iii) adapts the transition function on the basis of the executed actions and the effects observed via the perception function. The framework is able to deal with exogenous events that happen in the environment.

## 1 Introduction and Motivations

Automated Planning methods and techniques rely on models of the world, usually called Planning Domains. The (automated) acquisition of these models is widely recognised as a challenging bottleneck, see, e.g., the KEPS workshops and the ICKEPS competition.[1] The automated learning of planning domains is a way to address this challenge. Indeed, most often, it is impossible to specify a complete and correct model of the world. Moreover, most of the times a model needs to be updated and adapted to a changing environment.

Several and different learning approaches have been proposed so far. Some works on domain model acquisition focus on the problem of learning action schema, see, e.g. [Gregory and Cresswell, 2016; McCluskey *et al.*, 2009; Cresswell *et al.*, 2013; Mourão *et al.*, 2012; Mehta *et al.*, 2011; Zhuo and Yang, 2014]. Learning planning operators and domain models from plan examples and solution traces [Yang *et al.*, 2007; Zhuo *et al.*, 2010; Zhuo and Kambhampati, 2013;

---

[1]The Knowledge Engineering for Planinng and Scheduling (KEPS) Workshop and Competition (ICKEPS)

Henaff *et al.*, 2017] and learning probabilistic planning operators have also been investigated [Pasula *et al.*, 2004; Zettlemoyer *et al.*, 2005; Pasula *et al.*, 2007].

We propose a framework in which a discrete deterministic planning domain is extended with a perception function, i.e., a probabilistic mapping between state variables and observations from the real world represented by continuous variables, called perception variables. The perception function is represented by a conditional probability distribution that computes the likelihood of observing some values of the perception variables given an assignment to state variables.

We define an algorithm that builds an abstract deterministic finite planning domain and a perception function by executing actions and observing the effects through perception variables. The only information about the real world that is available to the learning algorithm is provided by the perceptions variables. The algorithm does not have access to a continuous model of the dynamics of the world. In several cases, such model is not available or is too difficult to provide.

The learning algorithm can start either "from scratch" (i.e., with an "empty planning domain"), or from some prior knowledge expressed with an initial discrete planning domain and perception function. The algorithm incrementally learns the values of the state variables, the description of the transition function, the constraints on state variables, and the perception function. The framework provides the ability to learn and adapt to unexpected situations, i.e., some constraints on state variables have been violated, or the domain of some state variables should be extended with new values.

The paper is structured as follows. Section 2 formalises the planning domain, including the perception function. Section 3 defines the incremental learning algorithm. In Section 4 we show how the algorithm works with an explanatory example that shows the potentialities of the framework. We finally discuss related work, conclusions, and future work.

## 2 Perceived Planning Domains

A *(deterministic) planning domain* is a triple $\mathcal{D} = \langle S, A, \gamma \rangle$, composed of a finite non empty set of states $S$, a finite non empty set of actions $A$, and a state transition function $\gamma : S \times A \rightarrow S$. Each state $s \in S$ is represented with a vector of *state variables* ranging over a finite set of values. Let $\boldsymbol{V} = \langle V_1, \ldots, V_m \rangle$ be a vector of $m$ state variables. Let $\boldsymbol{D} = \{D_1, \ldots, D_k\}$ be a set of non empty finite sets,

called *domains*. Let $\boldsymbol{Dom}$ be a function that assigns a domain $\boldsymbol{Dom}(V)$ to each variable $V$ of $\boldsymbol{V}$. The set $\boldsymbol{Dom}(V)$ is the set of values that can be assigned to the variable $V$. For every $\boldsymbol{W} \subseteq \boldsymbol{V}$, we use $\boldsymbol{Dom}(\boldsymbol{W})$ to denote the cross product of the domains of all the variables in $\boldsymbol{W}$, namely $\boldsymbol{Dom}(\boldsymbol{W}) = \underset{V \in \boldsymbol{W}}{\times} \boldsymbol{Dom}(V)$. For every $\boldsymbol{w} \in \boldsymbol{Dom}(\boldsymbol{W})$, we use $\boldsymbol{W} = \boldsymbol{w}$ to denote the *(partial) assignment* to each variable $V \in \boldsymbol{W}$ to $v \in \boldsymbol{Dom}(V)$. If $\boldsymbol{W}$ is the entire set of variables $\boldsymbol{V}$ then $\boldsymbol{V} = \boldsymbol{v}$ is a *total assignment*. A state $s \in S$ is a total assignment, i.e., a set of assignments that assigns a value $v \in \boldsymbol{Dom}(V)$ to every state variable $V$. We use $s[V]$ to denote the value assigned by $s$ to $V$. Not every total assignment necessarily corresponds to a state. The set of states $S$ of a planning domain is a subset of the total assignments. $S$ can be specified with a set of *constraints* between values of state variables. For instance, the fact that $V$ and $V'$ must take different values can be represented by the constraint $V \neq V'$. In this paper we suppose that constraints are expressed using propositional combination (via $\wedge$, $\vee$ and $\neg$) of the atomic proposition $V = v$, and $V = V'$, for $V, V' \in \boldsymbol{V}$ and $v \in \boldsymbol{Dom}(V)$.

We assume that the transition function $\gamma$ is specified with action language, resulting in a compact representation. In this paper we adopt a simple action language, which specifies $\gamma$ through a set of rules of the form

$$r : \; prec(r) \xrightarrow{a} \mathit{eff}(r) \tag{1}$$

where $a \in A$, $prec(r)$ is a propositional formula in the language of the constraints, and $\mathit{eff}(r)$ is a partial assignment $\boldsymbol{V}' = \boldsymbol{v}'$. For every action $a$ and state $s$, $s' = \gamma(a, s)$ is the state obtained after the execution of $a$ in $s$, and is defined as

$$s'[V] = \begin{cases} v & \text{if } \exists r \text{ for } a, \text{ such that } s \models prec(r) \text{ and} \\ & \quad \mathit{eff}(r) \text{ contains } V = v \\ s[V] & \text{otherwise} \end{cases}$$

In order to guarantee that $\gamma(a, s)$ is deterministic, we impose that for every pair of rules $r$ and $r'$, defining the action $a$, we have that if $prec(r) \wedge prec(r')$ is consistent then $\mathit{eff}(r) \cup \mathit{eff}(r')$ does not contain $V = v_1$ and $V = v_2$ for $v_1 \neq v_2$.

The agent perceives the world through a vector $\boldsymbol{X} = \langle X_1, \ldots, X_n \rangle$ of continuous variables ranging over real numbers, called *perception variables*. A *perception function*, is a function $f : \mathbb{R}^n \times \boldsymbol{Dom}(\boldsymbol{V}) \to R^+$, such that for every $\boldsymbol{x} \in \mathbb{R}^n$ and total assignment $\boldsymbol{V} = \boldsymbol{v}$, $f(\boldsymbol{x}, \boldsymbol{v}) = p(\boldsymbol{x} \mid \boldsymbol{V} = \boldsymbol{v})$, where $p(\boldsymbol{x} \mid \boldsymbol{V} = \boldsymbol{v})$ is a probability density funciton (PDF) that can be factorised as follows:

$$p(\boldsymbol{x} \mid \boldsymbol{V} = \boldsymbol{v}) = \prod_{i=1}^{n} p_{X_i}(x_i \mid \boldsymbol{V}_{J_i} = \boldsymbol{v}_{j_i})$$

where $\boldsymbol{V}_{J_i}$ is a subset of the state variables $\boldsymbol{V}$.

**Definition 1 (Extended planning domain)** *An extended planning domain is a pair $\langle \mathcal{D}, f \rangle$ where $\mathcal{D}$ is a planning domain and $f$ a perception function on the states of $\mathcal{D}$.*

Hereafter, if not explicitly specified, with "planning domain" we will refer to extended planning domain.

**Example 1 (The "Robot-Pack-Cat (RPC) Flat")** *The RPC-Flat is composed of 6 rooms (named from A to F), see Figure 1. In this flat there are a robot, a pack, and a cat. The robot can move from one room to adjacent rooms, load, transport and unload the pack. The cat moves around randomly and can also jump on top of the robot. The robot is equipped with an RFID reader able to perceive the presence in the room of the pack, which is equipped with a proximity sensor tag. Suppose that the robot has only partial knowl-*

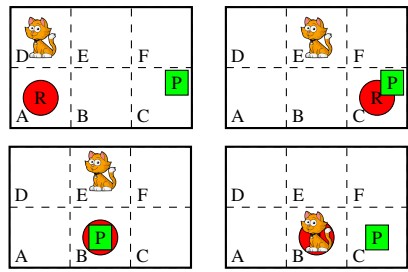

Figure 1: Four possible situations in the RPC-flat

*edge about the flat and its dynamics. It believes that there are only 4 rooms (ignoring the room C and F), it ignores also the presence of the cat. The robot represents its partial knowledge with the following planning domain: The states are represented by three state variables:* loc(r), loc(p), *and* loaded, *which represent the position of the robot, the position of the pack, and whether the robot is loaded. There are two domains i.e.,* $\boldsymbol{D} = \{$room, nr_of_carried_objects$\}$ *where* room $= \{0, 1, 2, 3\}$ *and* nr_of_carried_objects $= \{0, 1\}$, *with* $\boldsymbol{Dom}($loc(r)$) = $ room, $\boldsymbol{Dom}($loc(p)$) = $ room, *and* $\boldsymbol{Dom}($loaded$) = $ nr_of_carried_objects. *Notice that the robot assumes that there are only 4 rooms and 1 object to be carried.*

*Not all the state variable assignments are states (in $S$), indeed, when the robot is carrying the pack, their position must be the same. This can be formalized by the constraint:*

$$\mathsf{loaded} = 1 \to \mathsf{loc(r)} = \mathsf{loc(p)} \tag{2}$$

*The set $A$ of actions include* N, S, E, W *(that stand for the robot moves north, south, east, and west, respectively),* L, *and* U *(that stand for the robot loads and unloads the pack). Examples of a specification for* E *and* L *are the following:*

$$\mathsf{loc(r)} = 0 \xrightarrow{\mathsf{E}} \mathsf{loc(r)} = 1$$
$$\mathsf{loc(r)} = 0 \wedge \mathsf{loaded} = 1 \xrightarrow{\mathsf{E}} \mathsf{loc(p)} = 1$$
$$\mathsf{loc(r)} = \mathsf{loc(p)} \xrightarrow{\mathsf{L}} \mathsf{loaded} = 1$$

*The robot has the following perception variables:*

- *$X$, $Y$ with $\boldsymbol{Dom}(X) = \boldsymbol{Dom}(Y) = \mathbb{R}$ are the x- and y-coordinates of the position of the robot;*
- *$T$ with $\boldsymbol{Dom}(T) = [0, 1]$ is the output of RFID reader. If the pack and the robot are is in the same room then the value of $T$ is close to 1, otherwise it is close to 0;*
- *$W$ with $\boldsymbol{Dom}(W) = \mathbb{R}^+$ is the weight currently curried by the robot.*

*The perception function is factorized as follows:*

$$p(x, y, z, w \mid \mathsf{loc(r), loc(p), loaded}) = p_X(x \mid \mathsf{loc(r)}) \cdot$$
$$p_Y(y \mid \mathsf{loc(r)}) \cdot p_T(t \mid \mathsf{loc(r), loc(p)}) \cdot p_W(w \mid \mathsf{loaded}) \; where:$$

$$p_X(x \mid \mathsf{loc(r)}) = \mathcal{N}(x \mid \mu_{X, \mathsf{loc(r)}}, \sigma)$$

$$\mu_{X, \mathsf{loc(r)}} = \mathsf{loc(r)} \mod 2 + 0.5,$$

$$p_Y(y \mid \mathsf{loc(r)}) = \mathcal{N}(y \mid \mu_{Y, \mathsf{loc(r)}}, \sigma)$$

$$\mu_{Y, \mathsf{loc(r)}} = \mathsf{loc(r)} \div 2 + 0.5,$$

$$p_T(t \mid \mathsf{loc(r), loc(p)}) = \mathrm{B}(t \mid \alpha_{\mathsf{loc(r), loc(p)}}, \beta_{\mathsf{loc(r), loc(p)}})$$

$$\alpha_{\mathsf{loc(r), loc(p)}} = \cdot \mathbb{1}_{\mathsf{loc(r)=loc(p)}} + 2 \cdot \mathbb{1}_{\mathsf{loc(r) \neq loc(p)}}$$

$$\beta_{\mathsf{loc(r), loc(p)}} = 2 \cdot \mathbb{1}_{\mathsf{loc(r)=loc(p)}} + 1 \cdot \mathbb{1}_{\mathsf{loc(r) \neq loc(p)}}$$

$$p_W(w \mid \mathsf{loaded}) = \Gamma(w \mid k_{\mathsf{loaded}}, \theta_{\mathsf{loaded}})$$

$$k_{\mathsf{loaded}} = \mathsf{loaded} + 1, \; \theta = 1$$

## 3 The Incremental Learning Algorithm

The *Acting and Learning Planning-domains* algorithm, ALP, described in Algorithm 1, not only learns/updates the transitions of a planning domain, but it can also learn/update the perception function, and extend the set of states, either by weakening some constraints, or by extending the domains of some state variables. ALP can start "from scratch", i.e., from the simplest planning domain, where each variable domain $D \in \boldsymbol{D}$ is equal to $\{0\}$, without constraints, and an empty $\gamma$. Alternatively, ALP can start from any non empty planning domain corresponding to some "prior knowledge" about the world.

Given a planning domain with a set of state variables in input, ALP requires the perception function $f(\boldsymbol{x}, \boldsymbol{v}) = \prod_{i=1}^{n} p_{X_i}(\cdot | \boldsymbol{V}_{J_i} = \boldsymbol{v}_{J_i}))$ to be defined for all variable assignments $\boldsymbol{V} = \boldsymbol{v}$. Furthermore, since ALP introduces new values in the domain of state variables when the perception function of a perceived value $\boldsymbol{x}$ is too low, we need a method to intialise the perception function for these new values. For this reason ALP requires in input also an initialiser $p_{init, X_i}$, for every perception variable $X_i$, that returns a PDF for any observation $\boldsymbol{x}$. Moreover, ALP requires in input some additional *update parameters*, $\alpha$, $\beta$, $\gamma$, and $\delta$, all in [0,1], which determine how much the agent trusts in the various components of the model. In this section, we will explain the meaning of each parameter.

ALP iteratively refines the current planning domain $\mathcal{D}$ with the associated perception function $f$, by executing the actions proposed by EXPLORE (line 4),[2] and by observing the action effects through the perception variables $\boldsymbol{x}$ (line 5). In order to determine the next state $s_0'$ (from line 6 to line 15), ALP firstly computes ABOVETHRESHOLD$(\boldsymbol{x}, S)$ for the observation $\boldsymbol{x}$, which corresponds to the set of states such that the likelihood of observing each $x_i$ is above the threshold $(1-\epsilon) \cdot \max p_{X_i}$. Formally: ABOVETHRESHOLD$(\boldsymbol{x}, S)$ returns the set $\{s \in S \mid \forall i, \; p_{X_i}(x_i \mid s[\boldsymbol{V}_{J_i}]) \geqslant (1 - \epsilon) \cdot \max p_{X_i}\}$. Intuitively, ABOVETHRESHOLD selects a set of states that are

---

[2] A naïve implementation of EXPLORE can be a random generator of actions. A smarter strategy can take into account how much has the agent already learned, which portion of the domain has been already explored, and the part that still requires more learning.

---

**Algorithm 1** ALP

**Require:** $\mathcal{D} = \langle S, A, \gamma \rangle$ {Initial planning domain}
**Require:** $f = \prod p_{X_i}$ {Initial perception function}
**Require:** $s_0$ {Initial state}
**Require:** $\alpha, \beta, \delta, \epsilon$ {Update parameters}
**Require:** $p_{init, X_i}$ {Perception initialization for $X_i$}
**Require:** MAXITER {Maximum number of exploration steps}
1: $T \leftarrow \langle \rangle$ {The empty history of transitions}
2: $O \leftarrow \langle \rangle$ {The empty history of observations}
3: **for** ITER $\leftarrow 1$ **to** MAXITER **do**
4:     $a \leftarrow$ EXPLORE$(\mathcal{D}, s_0)$
5:     $\boldsymbol{x} \leftarrow$ ACT$(a)$
6:     $S_0' \leftarrow$ ABOVETHRESHOLD$(\boldsymbol{x}, S)$
7:     **if** $S_0' = \varnothing$ **then**
8:        $S_0' \leftarrow$ ABOVETHRESHOLD$(\boldsymbol{x}, \boldsymbol{Dom}(V) \backslash S)$
9:        **if** $S_0' = \varnothing$ **then**
10:           $\boldsymbol{D} \leftarrow$ EXTENDDOM$(\boldsymbol{D}, f, \boldsymbol{x})$
11:           $f \leftarrow$ EXTENDF$(\boldsymbol{D}, f, \boldsymbol{x})$
12:           $S_0' \leftarrow$ ABOVETHRESHOLD$(\boldsymbol{x}, \boldsymbol{Dom}_{new}(V))$
13:        **end if**
14:     **end if**
15:     $s_0' \leftarrow$ ONEOF$(\mathrm{argmax}_{s \in S_0'} f(\boldsymbol{x}, s) \cdot \mathrm{sim}(s, \gamma(s_0, a) \mid \delta))$
16:     **if** $s_0' \notin S$ **then**
17:        $S \leftarrow S \cup \{s_0'\}$
18:     **end if**
19:     $T \leftarrow$ APPEND$(T, \langle s_0, \pi(s_0), s_0' \rangle)$ {extend the transition history with the last one}
20:     $O \leftarrow$ APPEND$(O, \langle s_0', \boldsymbol{x} \rangle)$ {extend the observation history with the last one}
21:     $\gamma \leftarrow$ UPDATETRANS$(\gamma, T \mid \alpha)$
22:     $f \leftarrow$ UPDATEPERC$(f, O \mid \beta)$
23:     $s_0 \leftarrow s_0'$
24: **end for**

the candidates to be the next state, i.e., those states for which the likelihood of observing $x_i$ is higher than a certain threshold defined by the parameter $\epsilon \in [0, 1]$. At one extreme, when $\epsilon = 1$, ABOVETHRESHOLD$(\boldsymbol{x}, S)$ selects all states in $S$. On the other extreme, if $\epsilon = 0$, ABOVETHRESHOLD$(\boldsymbol{x}, S)$ selects only those states in which $f(\boldsymbol{x}, s)$ reaches its maximum value. The lower $\epsilon$, the higher chance to introduce new states. Intuitively, $\epsilon$ expresses how much we believe that the set of states learned so far are sufficient for the planning domain to model the real world.

At line 7, if there are no assignments among the current states that pass the threshold, then ALP considers the assignments which are not in the set of states, i.e., $\boldsymbol{Dom}(V) \backslash S$ (line 8). If, even in this case, ABOVETHRESHOLD returns the emtpy set (line 9), then we need to extend the possible assignments to variable by extending their domain. This is performed by EXTENDDOM (line 10), which extends the domain of one or more state variable.

EXTENDDOM (see Algorithm 2) takes in input the set $\boldsymbol{D}$ of current state variables domains, the perception function $f$, and the current observation $\boldsymbol{x}$. It starts by selecting one assignment $s$ that maximises the likelihood of observing $\boldsymbol{x}$. Then it computes the set $\boldsymbol{X}_{<lik}$ of perception variables $X_i$ where the likelihood of the perceived value $x_i$ w.r.t. the state $s$ is below the threshold (line 2). For every variable $X_i$ in $\boldsymbol{X}_{<lik}$, EXTENDDOM selects a domain in $\boldsymbol{D}_{J_i} = \{\boldsymbol{Dom}(V) \mid$

**Algorithm 2** EXTENDDOM

**Require:** $\boldsymbol{D}$ {The set of domain of state variables}
**Require:** $f = \prod p_{X_i}$ {Perception function}
**Require:** $\boldsymbol{x}$ {The result of a perception}
 1: $s \leftarrow \text{ONEOF}(\text{argmax}_{s \in \boldsymbol{Dom}(\boldsymbol{V})} f(\boldsymbol{x}, s))$
 2: $\boldsymbol{X}_{<lik} \leftarrow \{X_i \mid p_{X_i}(x_i \mid s[\boldsymbol{V}_{J_i}]) < (1 - \epsilon) \cdot \max p_{X_i}\}$
 3: $\boldsymbol{D}_H \leftarrow$ minimal hitting set of $\{\boldsymbol{D}_{J_i}\}_{X_i \in \boldsymbol{X}_{<lik}}$
 4: **for** $D \in \boldsymbol{D}_H$ **do**
 5: $\quad D \leftarrow D \cup \{|D|\}$
 6: **end for**
 7: **return** $\boldsymbol{D}$

---

**Algorithm 3** EXTENDF

**Require:** $\boldsymbol{D}$ {The set of domain of state variables}
**Require:** $\boldsymbol{x}$ {The result of a perception}
**Require:** $p_{init,X_i}$ {Perception initializator for $X_i$}
 1: **for** $\boldsymbol{v} \in \boldsymbol{Dom}(\boldsymbol{V})$ **do**
 2: $\quad$ **for** $X_i \in \boldsymbol{X}$ **do**
 3: $\quad\quad$ **if** $p_{X_i}(\cdot \mid \boldsymbol{V}_{J_i} = \boldsymbol{v}_{J_i})$ is not defined **then**
 4: $\quad\quad\quad p_{X_i}(\cdot \mid \boldsymbol{V}_{J_i} = \boldsymbol{v}_{J_i}) = p_{init,X_i}(x_i)$
 5: $\quad\quad$ **end if**
 6: $\quad$ **end for**
 7: **end for**

---

$V \in \boldsymbol{V}_{J_i}\}$ to be extended with a new value. Since we want to minimize the number of values introduced, we choose to extend the set of domains $\boldsymbol{D}_H$ that is a minimal hitting set[3] for $\{\boldsymbol{D}_{J_i}\}_{X_i \in \boldsymbol{X}_{<lik}}$ (line 3). Each domain in $D \in \boldsymbol{D}_H$ is extended with a new value, resulting in the set of $|D| + 1$ elements $\{0, 1, 2, \ldots, |D|\}$ (line 5).

After executing EXTENDDOM, ALP calls EXTENDF (line 11) to initialise the perception function for the newly introduced states. EXTENDF (see Algorithm 3) does this for all the variables without perception function (line 4). The introduction of the new values for state variables, and the initialisation guarantees that ABOVETHRESHOLD returns a non empty set $S'_0$ of assignments. Then ALP selects the next state $s'_0$ among the elements of $S'_0$ (line 15). The next state is one among the states that maximize the product of the likelihood of observing $\boldsymbol{x}$ and the similarity with the state predicted by the transition function learned so far, i.e., $\gamma(a, s_0)$. Ideally the next state will be the one that maximises the likelihood of the perceived values, and the closest to the state predicted by the model. These two sources of information however could be contradictory, therefore we have to jointly maximize their product.

The similarity/distance measure, $\text{sim}(s, s' \mid \delta)$ for $s, s' \in \boldsymbol{Dom}(\boldsymbol{V})$ is defined as

$$\prod_{i=1}^{m} \frac{1 + \delta \cdot (\mathbb{1}_{s[V_i]=s'(V_i)} \cdot (|\boldsymbol{Dom}(V_i)| - 1) - \mathbb{1}_{s[V_i]\neq s'(V_i)})}{1 + \delta(|\boldsymbol{Dom}(V_i)| - 1)}$$

The parameter $\delta \in [0, 1]$ allows us to adjust the similarity measure between states. At one extreme, if $\delta = 0$, then every

---

[3]A set of $A$ is an hitting set of a family of sets $\{B_i\}_{i=1}^{n}$ if $A \cap B_i \neq \varnothing$ for every $i$. $A$ is a minimal hitting set if there is no hitting set $A'$ for $\{B_i\}_{i=1}^{n}$ with $|A'| < |A|$.

---

state is similar to every other state, i.e., $\text{sim}(s, s' \mid \delta) = 1$, and the similarity does not play any role in the maximisation. If $\delta = 1$, sim coincides with the equality relation, i.e., $\text{sim}(s, s' \mid \delta) = \mathbb{1}_{s=s'}$, which implies that the maximization will always return $\gamma(a, s_0)$. The interesting case is when $\delta \in (0, 1)$. The lower $\delta$, the more we trust in the perceptions of the agent's sensors. The higher $\delta$, the more we trust in the model learned so far.

If $s'_0$ is not part of the current set of states $S$, we have to include it by weakening the constraints. Let $C_1, \ldots, C_k$ be the set of constraints defining $S$. To specify $S \cup \{s'_0\}$, we have to weaken each $C_i$ as follows

$$C_i \vee \bigwedge_{V \in \boldsymbol{V}} V = s'_0[V] \quad (3)$$

and if the new values $v_{new}$ is introduced we have to add the following constraint:

$$V = v_{new} \rightarrow \bigwedge_{V' \neq V} V' = s'_0[V'] \quad (4)$$

for every variable $V$ for which the domain $\boldsymbol{Dom}(V)$ has been extended with the new value $v_{new}$.

**Proposition 3.1** *Let* $\{C'_i\}_{i=1}^{h}$ *be the set of constraints resulting from the revision of* $\{C_i\}_{i=1}^{k}$ *according to the rules* (3) *and* (4), *then* $s \models \bigwedge_{i=1}^{h} C'_i$ *if and only if* $s \in S \cup \{s'_0\}$.

ALP then extends the sequence of transitions $T$ and of observations $O$, and learn the new transition function $\gamma$ and the new perception function $f$. The functions UPDATETRANS and UPDATEPERC update the transition function $\gamma$ and the perception function $f$, respectively, depending on the data available in $T$ and $O$. The update functions take into account (i) the current model, (ii) what has been observed in the past, i.e., $T$ and $O$, and (iii) what has been just observed, i.e., $\langle s_0, a, s'_0 \rangle$ and $\langle s'_0, \boldsymbol{x} \rangle$. The update functions can be defined in several different ways, depending on whether we follow a cautious strategy, where changes are made only if there is a certain number of evidences from acting and perceiving the real world, or a more impulsive reaction to what the agent has just observed. In the following, we describe in detail how we create/update transitions, and how we create/update perception functions.

**Updating transitions.** UPDATETRANS decides whether and how to update the transition function. If $s'_0$ is the state that maximises the product of the perception function and of the similarity, and $s'_0$ is different from the state predicted by the planning domain, i.e., $s'_0 \neq \gamma(a, s_0)$, then $\gamma$ may need to be revised to take into account this discrepancy. Since our domain is deterministic (the transition $\gamma$ must lead to a single state), if the execution of an action leads to an unexpected state, we have only two options: either change $\gamma$ with the new transition or not. We propose the following transition update function that depends on $\alpha$: We define UPDATETRANS$(\gamma, T)(s, a) = s'$ where $s'$ is a state that maximizes

$$\alpha \cdot \mathbb{1}_{s'=\gamma(s,a)} + (1 - \alpha) \cdot |\{i \mid T_i = \langle s, a, s' \rangle\}| \quad (5)$$

where $T_i$ is the $i$-th element of $T$, and $\alpha \in [0, 1]$. Notice that, if $\alpha = 1$, we are extremely cautious, we strongly believe in our model of the world, and we never change the transition $\gamma$. Conversely, if $\alpha = 0$, we are extremely impulsive,

we do not trust our model, and just one evidence makes us to change the model. In the intermediate cases, $\alpha \in (0, 1)$, depending on the value of $\alpha$, we need more or less evidence to change the planning domain. In order to update $\gamma$, we have to revise the action specifications. We replace every rule $r$ about $a$ of the form $r : prec(r) \xrightarrow{a} \mathit{eff}(r)$, such that $s \models prec(r)$ and not $s \models \mathit{eff}(r)$ with the following rules for every $V_i$

$$r'_i : prem(r) \wedge V_i \neq s[V_i] \xrightarrow{a} \mathit{eff}(r)$$

and the following rule for all $j$, such that $s[V_j] \neq s'[V_j]$

$$r''_j : \bigwedge_{j=1}^{m} V_j = s[V_j] \xrightarrow{a} V_j = s'[V_j]$$

Notice that this method might generate a proliferation of very specific rules. Therefore after this step it is convenient to apply some algorithm for rule factorisation. Examples of factorization rules are the following:

$$\begin{matrix} \Gamma, V = v \xrightarrow{a} V' = v' \\ \Gamma, V \neq v \xrightarrow{a} V' = v' \end{matrix} \text{ are merged in } \Gamma \xrightarrow{a} V' = v'$$

Another example, is the following. Suppose that $\boldsymbol{Dom}(V) = \{0, 1, 2\}$, then:

$$\begin{matrix} \Gamma, V = 0 \xrightarrow{a} V' = v' \\ \Gamma, V = 1 \xrightarrow{a} V' = v' \end{matrix} \text{ are merged in } \Gamma, V \neq 2 \xrightarrow{a} V' = v'$$

A final example is the following. Suppose that $\boldsymbol{Dom}(V)$ and $\boldsymbol{Dom}(V')$ are equal to $\{0, 1\}$ then

$$\begin{matrix} \Gamma, V = 0, V' = 0 \xrightarrow{a} V'' = v'' \\ \Gamma, V = 1, V' = 1 \xrightarrow{a} V'' = v'' \end{matrix} \text{ are merged in}$$

$$\Gamma, V = V' \xrightarrow{a} V'' = v''$$

Dealing with rule factorisation can be considered as a separate topic and for lack of space is not treated in this paper. However, this operation results crucial in order to generate compact and "understandable" description of the transition function.

**Updating the perception function.** The update of the perception function is based on the current perception function $f(\boldsymbol{x}, s)$ for $s \in S$ and the set of observations $O$. We suppose that each function $p_{X_i}$ composing the perception function $f = \prod_i p_{X_i}$, belongs to a parametric family with parameters $\boldsymbol{\theta}_{X_i}$. For every partial assignment $\boldsymbol{V}_{J_i} = \boldsymbol{v}_{J_i}$ to the state variables $\boldsymbol{V}_{J_i}$ from which $X_i$ depends on, $p_{X_i}(\cdot \mid \boldsymbol{v}_{J_i})$ is obtained by setting the parameters $\boldsymbol{\theta}_{X_i}$ to some value $\boldsymbol{\theta}_{X_i, \boldsymbol{v}_{J_i}}$. In Example 1, $p_X$ is a Gaussian distribution with parameters $\boldsymbol{\theta}_X = \langle \mu_X, \sigma_X \rangle$. For every value $r \in \boldsymbol{Dom}(\mathsf{loc}(\mathsf{r}))$, $\mu_{X,r}$ and $\sigma_{X,r}$ are the mean and the standard deviation of $p_X$ and $p_X(x \mid \mu_{X,r}, \sigma_{X,r}) = \mathcal{N}(x, \mu = \mu_{X,r}, \sigma = \sigma_{X,r})$ expresses the likelihood of observing $x$ when the robot is in the room $r$. We denote by $\boldsymbol{\theta}_X$ all the parameters in $\boldsymbol{\theta}_{X_1}, \dots, \boldsymbol{\theta}_{X_n}$, and $\boldsymbol{\theta}_{X,\boldsymbol{v}}$, for $\boldsymbol{v} \in \boldsymbol{Dom}(\boldsymbol{V})$, their instantiations $\boldsymbol{\theta}_{X_1, \boldsymbol{v}_{J_1}}, \dots, \boldsymbol{\theta}_{X_n, \boldsymbol{v}_{J_n}}$.

Given a set of observations about the state $s$, $O(s) = \langle \boldsymbol{x}^{(0)}, \dots, \boldsymbol{x}^{(k)} \rangle$ and a new observation $\langle \boldsymbol{x}^{(k+1)}, s \rangle$ we have to update the values of $\boldsymbol{\theta}_{X,s}$ in order to maximise a combination of the current belief of the agent and the likelihood of

the entire set of observations extended with the new observation. Also in this case the agent can be more or less careful in the revision, being more or less confident in its beliefs. The update equation is therefore defined as:

$$\boldsymbol{\theta}'_{\boldsymbol{X},s} = \beta \cdot \boldsymbol{\theta}_{\boldsymbol{X},s} + (1 - \beta) \cdot \underset{\boldsymbol{\theta}'}{\operatorname{argmax}} \mathcal{L}(\boldsymbol{\theta}', \boldsymbol{x}^{(i)}, \dots, \boldsymbol{x}^{(k+1)}, s)$$

where the parameter $\beta \in [0, 1]$, expresses agents's confidence in its beliefs; the higher the value of $\beta$ the more careful the agent is in the revision, and

$$\mathcal{L}(\boldsymbol{\theta}, \boldsymbol{x}^{(1)}, \dots, \boldsymbol{x}^{(k)}, s) \propto \prod_{j=1}^{k} f(\boldsymbol{x}^{(j)}, s)$$

Due to the factorization of the perception function $f = \prod p_{X_i}$, we can separately update each set of parameters $\boldsymbol{\theta}_{X_i}$ associated to the perception variable $X_i$, defining therefore

$$\boldsymbol{\theta}_{X_i,s}^{k+1} = \beta \cdot \boldsymbol{\theta}_{X_i,s}^{k} + (1 - \beta) \cdot \underset{\boldsymbol{\theta}'_{X_i}}{\operatorname{argmax}} \prod_{j=1}^{k} p_{X_i}(x_i^{(j)} \mid \boldsymbol{\theta}'_{X_i})$$

(6)

## 4 Explanatory Examples

Let us now show how ALP works in Example 1. In Example 2, we first show how ALP learns new states by extending the domain of state variables and weakening constraints. In Example 3, we show how ALP can deal with highly unexpected events by adapting the planning domain.

**Example 2** *Let us suppose that the robot starts with the planning domain described in Example 1.*

*1. Suppose that the robot believes to be in the state $s_0$ where $\mathsf{loc}(\mathsf{r}) = 0$, $\mathsf{loc}(\mathsf{p}) = 1$, and $\mathsf{loaded} = 0$ (shortly written as $s_0 = 010$), and that the world is in the state shown in the top-left rectangle of Figure 1. Suppose that EXPLORE generates the action $\mathsf{E}$ (line 4) and the execution of this action moves the robot of about one unit in the east direction. The observation returned after the execution (line 5) is $\boldsymbol{x} = \langle x, y, t, w \rangle$ with $x \approx 1.5$, because the robot is moving east of approximately 1 unit; $y \approx 0.5$, because the robot is moving approximately horizontally; $t \approx 0$, because, differently from the model, the pack is not in that room; finally $w \approx 0$, since the robot is carrying nothing.*

*2. ALP computes the set of states $S'_0 \subseteq S$ that are above the threshold (line 6). Let us suppose that $\epsilon = 0.5$, i.e., we decide to balance our trust in the initial set of states and in the perceptions after executing actions. The robot position perception variables $x$ and $y$ indicate that the robot is in room 1 ($\mathsf{loc}(\mathsf{r}) = 1$). The sensor tag perception variable $t$ indicates that the pack is not in the same room of the robot, i.e., $\mathsf{loc}(\mathsf{p})$ is equal to 0, or 2, or 3. The weight perception variable $w$ indicates that the robot is not loaded, i.e., $\mathsf{loaded} = 0$. Therefore, $S'_0 = \{100, 120, 130\}$.*

*3. Since $S'_0 \neq \emptyset$, ALP computes the set $S'_0$ of the states in $S'_0$ that maximise $f(x, y, t, w, s) \cdot \mathrm{sim}(s, 110 \mid \delta)$. (line 15). Notice that $\gamma(\mathsf{E}, s_0) = \gamma(\mathsf{E}, 010) = 110$. Notice that*

in all the states $s \in S_0'$, $f(x, y, t, w, s)$ is the same, and it approximately equal to

$$\mathcal{N}(1.5 \mid \mu = 1.5, \sigma = 1) \cdot \mathcal{N}(0.5 \mid \mu = 0.5, \sigma = 1)$$
$$\cdot \, B(0 \mid \alpha = 1, \beta = 2) \cdot \Gamma(0 \mid k = 0, \theta = 1)$$

*i.e., the robot is unloaded and in room 1, and the pack is in a different room. The values of the factor $\text{sim}(s, 110 \mid \delta)$ is also the same for all the elements of $S_0'$. Therefore* ALP *randomly select one state of $S_0'$. Suppose that* ALP *selects $s_0' = 130$.*

*4. Then* ALP *jumps to line 19 and the transition $\langle 010, \mathsf{E}, 130 \rangle$ is added to the transition log $T$, and the observation $\langle 130, \boldsymbol{x} \rangle$, with $\boldsymbol{x} \approx \langle 1.5, 0.5, 0, 0 \rangle$ is added to the observation log $O$ (line 20).*

*5. Then* ALP *revises the transition function $\gamma$ (line 21). According to equation (5), with $a = \mathsf{E}$ and $s = 010$, we have:*

| $s'$ | equation (5) | $\alpha = 0$ | $\alpha = \frac{1}{2}$ | $\alpha = 1$ |
|------|------|------|------|------|
| $130$ | $\alpha \cdot 0 + (1 - \alpha) \cdot 1$ | $1$ | $\frac{1}{2}$ | $0$ |
| $110$ | $\alpha \cdot 1 + (1 - \alpha) \cdot 0$ | $0$ | $\frac{1}{2}$ | $1$ |
| *others* | $\alpha \cdot 0 + (1 - \alpha) \cdot 0$ | $0$ | $0$ | $0$ |

*If $\alpha > 1/2$ then $\gamma$ will not be changed, otherwise $\gamma(010, \mathsf{E}) = 130$. Let us suppose that $\alpha > 1/2$.*

*6. The new current state $s_0$ is set to $130$ (line 23), and a new action is generated by* EXPLORE *(line 4). Let's suppose it is again* E*. The values returned by the perception function are $x \approx 2.5$ and $y \approx 0.5$, since the action* E *moves the robot east of one unit (this is possible since actually there is a room east of room 1); $t \approx 1$ (since now the pack is actually in the same room of the robot), and $w \approx 0$ (since the robot is unloaded).*

*7. Now there are no states in $S$ that are above the threshold (line 6), since $p_X(2.5 \mid s)$ is very low for all the states $s \in S$. Therefore $S_0' = \varnothing$.*

*8.* ALP *checks therefore if there are assignments to state variables that are not states in $S$ that have the perception function above the threshold (line 8). Even in this case, for the same reason, no assignment allows for a perception function that is above the threshold. Therefore $S_0'$ is again empty.*

*9.* ALP *generates therefore a new state by extending the domain of state variables (line 10).* EXTENDDOM *starts by computing the states that maximizes the likelihood of observing $\langle x, y, t, w \rangle \approx \langle 2.5, 0.5, 1, 0 \rangle$, i.e., $s = 110$. Notice that $P_Y(\approx 0.5 \mid \text{loc}(\mathsf{r}) = 1)$ is close to the maximum of $P_Y$; similarly for $P_T(\approx 1 \mid \text{loc}(\mathsf{r}) = \text{loc}(\mathsf{p}))$ and $P_W(\approx 0 \mid \text{loaded} = 0)$ are also close to the maximum of $P_T$ and $P_W$ respectively. So if $\epsilon$ is small enough (i.e., the robot is enough "open" to the introduction of new states), $X$ is the only variable for which $P_X(\approx 2.5 \mid \text{loc}(\mathsf{r}) = 1)$ is below the threshold. Therefore $\boldsymbol{X}_{\leqslant lik} = \{X\}$, and $\boldsymbol{D}_H = \{\text{room}\}$ (line 3 of algorithm* EXTENDDOM*).*

*10. The domain* room *is therefore extended with a new value, obtaining* room $= \{0, 1, 2, 3, 4\}$*. Notice that, since* room *is also the domain of the variable* $\text{loc}(\mathsf{p})$*, this implies that we also extend the domain of this variable. With this extension we pass from $4 \cdot 4 \cdot 2 = 32$ possible assignments to $5 \cdot 5 \cdot 2 = 50$ possible assignments.*

*11.* EXTENDF *(line 11) extends the perception function for the new assignments: $p_X(x \mid \text{loc}(\mathsf{r}) = 4) = \mathcal{N}(x \mid \mu_{\text{loc}(\mathsf{r})=4} \approx 2.5, \sigma = 1)$, and $p_Y(y \mid \text{loc}(\mathsf{r}) = 4) = \mathcal{N}(x \mid \mu_{\text{loc}(\mathsf{r})=4} \approx 0.5, \sigma)$. $p_T(t \mid s)$ when $s$ contains the new value 4 is already defined, and the perception function for $W$ is not extended since it is not related to the state variable with domain* room*.*

*12. The $s_0'$ that maximises the new perception function is then $440$ (lines 15 and 15).* ALP *therefore updates the constraints in order to include only $440$ as a new state. According to Formula (4),* ALP *generates the following new constraints:*

$$\text{loc}(\mathsf{r}) = 4 \to \text{loc}(\mathsf{p}) = 4 \land \text{loaded} = 0 \qquad (7)$$
$$\text{loc}(\mathsf{p}) = 4 \to \text{loc}(\mathsf{r}) = 4 \land \text{loaded} = 0 \qquad (8)$$

*According to Formula (3),* ALP *updates the previous constraint as follows:*

$$(\text{loaded} = 1 \to \text{loc}(\mathsf{r}) = \text{loc}(\mathsf{p})) \lor$$
$$(\text{loc}(\mathsf{r}) = 4 \land \text{loc}(\mathsf{p}) = 4 \land \text{loaded} = 0)$$

*which is equivalent to* $\text{loaded} = 1 \to \text{loc}(\mathsf{r}) = \text{loc}(\mathsf{p})$*. Therefore* ALP *adds only the constraints (7) and (8).*

*13. $T$ becomes $\langle \langle 010, \mathsf{E}, 130 \rangle, \langle 130, \mathsf{E}, 440 \rangle \rangle$; $O$ becomes approximately the list of $\langle \approx \langle 0.5, 0.5, 0, 0 \rangle, 010 \rangle$, $\langle \approx \langle 1.5, 0.5, 0, 0 \rangle, 130 \rangle$, and $\langle \approx \langle 2.5, 0.5, 1, 0 \rangle, 440 \rangle$.*

*14. Suppose that the parameters $\alpha$ and $\beta$ are high enough not to affect the change of $\gamma$ and that they have a minimal effect on the perception function. The new state $s_0$ is now set to $440$.*

*15. Suppose that* EXPLORE *returns action load,* L*. The new perceived values are approximately $\langle x, y, t, w \rangle \approx \langle 2.5, 0.5, 1, 1 \rangle$. None of the states in $S$ is such that $p_W(w \mid s)$ is above the threshold (line 6).* ALP *checks therefore if there is some assignment that does not satisfy the constraints with a better likelihood (line 8). Indeed the assignment $441$ is such that all the likelihoods $p_X(2.5 \mid \text{loc}(\mathsf{r}) = 4)$, $p_Y(0.5 \mid \text{loc}(\mathsf{r}) = 4)$, $p_T(1 \mid \text{loc}(\mathsf{r}) = 4, \text{loc}(\mathsf{p}) = 4)$, and $p_W(1 \mid \text{loaded} = 1)$ are above the threshold. This meas that $s_0' = 441$ is the new state, and* ALP *adds it to $S$ (line 17). Adding $441$ to the set of states amounts to revise the constrains following formula (3). After some simplification,* ALP *obtains the constraints*

$$\text{loc}(\mathsf{r}) = 4 \to \text{loc}(\mathsf{p}) = 4 \qquad (9)$$
$$\text{loc}(\mathsf{p}) = 4 \to \text{loc}(\mathsf{r}) = 4 \qquad (10)$$
$$(\text{loaded} = 1 \to \text{loc}(\mathsf{r}) = \text{loc}(\mathsf{p})) \qquad (11)$$

**Example 3** *We continue the previous example showing how* ALP *adapts the planning domain to unexpected situations.*

*1. Suppose that* EXPLORE *returns the action* W *(go west) and that while executing this action the pack unexpectedly falls down and remains in room $C$, and simultaneously the cat (with a similar weight of the pack) jumps on top of the robot (!) (see bottom-right rectangle of Figure 1).*

*2. The sensors return $\boldsymbol{x} \approx \langle 1.5, 0.5, 0, 1 \rangle$. $f(\boldsymbol{x}, s)$ is very low (below the threshold) for all the states in $S$ since $w \approx 1$ should mean that the pack is loaded, while $t \approx 0$ tells us that the pack is not in the same room of the robot, and the constraint (2) imposes that $\text{loc}(\mathsf{r}) = \text{loc}(\mathsf{p})$.*

*3.* ALP *now checks the assignments to state variables that are not states in $S$ (line 8). The assignment corresponding to the actual situation, i.e., the robot is loaded and in a different room from the pack, that is* 141*, maximises* $f(\boldsymbol{x}, s) \cdot \text{sim}(s, \gamma(441, \mathsf{W}) \mid \delta)$ *(with $\delta < 1$). To extend $S$ with* 141 *(line 17),* ALP *weakens the constraints according to rules* (3) *and* (4) *obtaining:*

$\mathsf{loc}(\mathsf{r}) = 4 \rightarrow \mathsf{loc}(\mathsf{p}) = 4$

$\mathsf{loc}(\mathsf{p}) = 4 \rightarrow \mathsf{loc}(\mathsf{r}) = 4 \vee (\mathsf{loc}(\mathsf{r}) = 1 \wedge \mathsf{loaded} = 1)$

$\mathsf{loaded} = 1 \rightarrow \mathsf{loc}(\mathsf{r}) = \mathsf{loc}(\mathsf{p}) \vee (\mathsf{loc}(\mathsf{r}) = 1 \wedge \mathsf{loc}(\mathsf{p}) = 4)$

*4. Finally, suppose that, while the robot is carrying the pack, the cat jumps on top of the pack. The perception variable $W$ will return a value around 2 (1 for the pack plus 1 for the cat) and $p_W(w \mid \mathsf{loaded})$ will be below the threshold for all the states $s \in S$. Then* ALP *will extend the domain of the Boolean variable* loaded*, which becomes a three-valued variable, i.e.,* nr_of_carried_objects *is extended from* $\{0, 1\}$ *to* $\{0, 1, 2\}$.

## 5 Related Work

As far as we know, the problem addressed in this paper is novel, as well as the approach and the proposed solution. Some works on domain model acquisition focus on the problem of learning action schema from collections of plans, see, e.g. [Gregory and Cresswell, 2016; McCluskey *et al.*, 2009; Cresswell *et al.*, 2013; Mourão *et al.*, 2012; Mehta *et al.*, 2011; Zhuo and Yang, 2014]. They do not consider perceptions and the set of states is given.

Works on learning and planning in POMDP (see, e.g., [Ross *et al.*, 2011; Katt *et al.*, 2017]) learn a model of the POMDP domain through interactions with the environment, with the goal to do planning, e.g., by reinforcement learning or by sampling methods. They learn the transitions, while the set of states is given as well as the mapping through observations.

Some works on POMDP Model Learning, see, e.g., [van Otterlo, 2009; Zheng *et al.*, 2018], drop the assumption that the set of states is given or the bound on the number of states is known. Two main differences with our work still exist. First, we do not learn a POMDP model, we learn a deterministic model that enables efficient planning techniques. Second, we learn the set of states represented through state variables and constraints, which is the practical way to represent a planning domain.

Our approach shares some similarities with the work on planning by reinforcement learning [Kaelbling *et al.*, 1996; Sutton and Barto, 1998; Geffner and Bonet, 2013; Yang *et al.*, 2018; Parr and Russell, 1997; Ryan, 2002; Leonetti *et al.*, 2016], since we learn by acting in the environment. However, these works focus on learning policies and assume the set of states and the correspondence between continuous data from sensors and states are fixed.

Different approaches are those followed by LatPlan and Causal InfoGAN. Causal InfoGAN [Kurutach *et al.*, 2018] learns discrete or continuous models from high dimensional sequential observations. This approach fixes a priori the size of the discrete domain model, and performs the learning off line. Differently from our approach their goal is to generate an execution trace in the high dimensional space. LatPlan [Asai and Fukunaga, 2018] takes in input pairs of high dimensional raw data (e.g., images) corresponding to transitions. It also takes an offline approach. Our approach is online and local, we can therefore deal with a dynamic environment.

A complementary approach is pursued in works that plan and learn directly in a continuous space, see e.g., [Abbeel *et al.*, 2006; Mnih *et al.*, 2015; Co-Reyes *et al.*, 2018]. These approaches do not require a perception function, since there is no abstract discrete model of the world. Such approaches are very suited to address some tasks, e.g., moving a robot arm to a desired position or performing some manipulations. However, we believe that, in several situations, it is conceptually appropriate and practically efficient to learn an abstract discrete and deterministic model where planing is much easier and efficient to perform.

Finally, we share the idea of a planning domain at the abstract level with all the work on abstraction on MDP models, see, e.g., [Abel *et al.*, 2018]. However, our problem and approach is substantially different, since in the work on abstraction on MDP models the mapping between the original MDP and the abstract states is given, while we learn it.

## 6 Conclusion and Future Work

We believe this work opens a new perspective in learning planning domains and perceptions through continuous observations. The framework provides the ability to learn domains represented with state variables and constraints, which is the natural way to represent planning domains. Learning a finite deterministic planning domain represented with state variables opens up the possibility to use all the available efficient planners to reason at the abstract level. Learning the perception function takes into account the fact that, while an agent can conveniently plan at the abstract level, it perceives the world and acts through sensors and actuators that work in a continuous space. Learning perception functions allows us to learn new states that represent unexpected situations of the world. Finally, the framework allows us to learn domains incrementally, and to adapt to a changing environment.

Still a lot of work remains to do. A proof of convergence to coherent models should be provided, and the conditions of convergence should be defined. The framework should be implemented and an experimental evaluation should be performed. Additional work needs to be done to support more sophisticated action and constraint revisions on the basis of the observed transition. Finally, the ALP algorithm should be integrated with a state-of-the-art on-line planner and with efficient exploration techniques.

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
