# OpenReview forum: "Incremental Learning of Discrete Planning Domains from Continuous Perceptions"
_icaps-conference.org/ICAPS/2019/Workshop/KEPS — KEPS 2019_

### Official Review · AnonReviewer1 · 2019-04-26
**Acquisition of discrete models in continuous scenarios**

**Rating:** 4
**Confidence:** 3

**Review:**

This work introduces a framework for learning discrete domain models and a perception function in an iterative fashion, by executing actions and observing the resulting effects via perception variables. The proposed algorithm relies on the information provided by perception variables to generate a domain model form scratches (or updating an existing one).

The paper correctly position itself with regards to the relevant literature. Differently from existing works, it does not consider partial observability cases, and assumes that it is possible to freely and safely explore the environment. The latter, in particular, is a quite strong assumption, but it is needed in order to allow the framework to deal with the newly-introduced perception variables. However, from a practical perspective, I'm wondering how often that will be allowed in real-world applications.

The proposed approach (summarised in Algorithm 1) involves the exploration of the possible actions, in order to observe results and update relevant aspects of the extended domain model. The involved steps are well defined and easy to follow. I also appreciated the use of numerous examples, to guide the reader and better clarify how the proposed framework works.

As a comment for future extensions, it would be interesting to test how the proposed algorithm performs in practice, and what aspects of the application domain affect the quality of the generated final domain model. I'm also wondering whether this approach could be used in combination with existing works: other approaches provide a first domain model that ignores numerical aspects, and ALP extends them to handle them. It would also be great to evaluate whether ALP can be a suitable tool for cooperation between human and machine in generating domain models.

- Latex Style comment: Please make sure to use the AAAI style.

---

### Official Review · AnonReviewer2 · 2019-05-03
**Extremely relevant paper, with a lot of potential**

**Rating:** 4
**Confidence:** 3

**Review:**

The paper focuses on a theoretical framework for inducing a discrete planning domain in a very general sense, from the perspective of an agent that can act in an environment and receive numeric sensor data.

Perhaps the most attractive aspect of this paper is the ambitious generality of the model learning: it's not just filling in templated action schemas, but going to the point of having an unbounded model capacity a priori (where new values for a discrete domain can be dynamically learned). As far as I can tell, the set of actions and variables are fixed, but what is presented is a nice and important step for the general setting of learning abstract action representations from noisy and continuous environments.

The work is directly aligned with the special focus of KEPS this year, and my recommendation is acceptance. That said, I think there are a number of areas for improvement that the authors should consider.

* One major missing piece of related work is that of Konidaris et al.
- http://lis.csail.mit.edu/pubs/konidaris-jair18.pdf
- Additionally, contrasting things with the model reconciliation work would be useful, especially in the context of modifications to the action theory.

* It took a while to understand that "states for the system" in fact meant the valid reachable states (through the specification of invariants). This should be clarified. Perhaps it is not strictly the set of reachable states, but at least it represents a subset of the full state space represented by the product of variables (which is typically what is meant by "all states for a domain").

* I found the details on the perception functions extremely hard to parse (e.g., the definitions just before section 3). Some added focus here would help with clarity.

* There are a number of parameters you use, and their descriptions are scattered throughout the algorithm text. It would help to have them in a single table with the intuitive explanations listed (e.g., "\eps expresses how much we believe that the set of states learned so far are sufficient for the planning domain to model the real world"). These intuitive explanations are really enlightening, and help the reader to understand the formulae they appear in -- put them together in a central location to surface these important points.

* Small error: top of page 5: "prem" should be "prec"

* It seems as though errors will cascade as part of the system: you incorporate the observations in a 1-step fashion. Conversely, if you were to look at an entire episode, or a longer sequence of actions, you should be able to define an update that maximizes the posterior likelihood across the entire sequence. There are obvious connections to RL, and in particular you would have a nice Bayesian analogy to TD(n) learning for your model updates (and subsequently can discuss the tradeoff between accuracy from more history and the computational cost it comes with).

* The examples (especially the third) seem to be a happy path coincidence (your parameters happen to be set the right way, etc). But it's worth also pointing out a sad path for this work -- when do things fail and cascade to a horribly learned model? I don't believe you are protected from such failures (especially with the myopic view of the updates), and it would help with understanding to know both how it works as well as how it fails.

* I realized that the aim is to learn a deterministic model, but even the motivating example is inherently stochastic (with other agents -- i.e., cats -- messing with the system dynamics). A discussion about issues that arise here would be interesting. Note that a mostly deterministic environment that changes is characteristically different than a stochastic environment. In the former, you can recognize when things have shifted (and perhaps dynamically adjust the 4 parameters in response), but in the latter you will constantly thrash between deciding the one true outcome of actions.